# Sex-, age-, and organ-dependent improvement of bile acid hydrophobicity by ursodeoxycholic acid treatment: A study using a mouse model with human-like bile acid composition

Hajime Ueda[1], Akira Honda[1,2]*, Teruo Miyazaki[2], Yukio Morishita[3], Takeshi Hirayama[1], Junichi Iwamoto[1], Nobuhiro Nakamoto[4], Tadashi Ikegami[1]

1 Division of Gastroenterology and Hepatology, Tokyo Medical University Ibaraki Medical Center, Ibaraki, Japan, 2 Joint Research Center, Tokyo Medical University Ibaraki Medical Center, Ibaraki, Japan, 3 Diagnostic Pathology Division, Tokyo Medical University Ibaraki Medical Center, Ibaraki, Japan, 4 Department of Internal Medicine, Division of Gastroenterology and Hepatology, Keio University School of Medicine, Tokyo, Japan

* akihonda@tokyo-med.ac.jp

**Data Availability Statement:** All relevant data are within the manuscript and its Supporting Information files.

## Abstract

$Cyp2a12^{-/-}Cyp2c70^{-/-}$ double knockout (DKO) mice have a human-like hydrophobic bile acid (BA) composition and show reduced fertility and liver injury. Ursodeoxycholic acid (UDCA) is a hydrophilic and cytoprotective BA used to treat various liver injuries in humans. This study investigated the effects of orally administered UDCA on fertility and liver injury in DKO mice. UDCA treatment prevented abnormal delivery (miscarriage and preterm birth) in pregnant DKO mice, presumably by increasing the hydrophilicity of serum BAs. UDCA also prevented liver damage in six-week-old DKO mice, however liver injury emerged in UDCA-treated 20-week-old female, but not male, DKO mice. In 20-week-old male UDCA-treated DKO mice, conjugated plus unconjugated UDCA proportions in serum, liver, and bile were 71, 64, and 71% of the total BAs, respectively. In contrast, conjugated plus unconjugated UDCA proportions in serum, liver, and bile of females were 56, 34, and 58% of the total BAs, respectively. The UDCA proportion was considerably low in female liver only and was compensated by highly hydrophobic lithocholic acid (LCA). Therefore, UDCA treatment markedly reduced the BA hydrophobicity index in the male liver but not in females. This appears to be why UDCA treatment causes liver injury in 20-week-old female mice. To explore the cause of LCA accumulation in the female liver, we evaluated the hepatic activity of CYP3A11 and SULT2A1, which metabolize LCAs to more hydrophilic BAs. However, there was no evidence to suggest that either enzyme activity was lower in females than in males. As female mice have a larger BA pool than males, excessive loading of LCAs on the hepatic bile salt export pump (BSEP) may be the reason for the hepatic accumulation of LCAs in female DKO mice with prolonged UDCA treatment. Our results suggest that the improvement of BA hydrophobicity in DKO mice by UDCA administration is sex-, age-, and organ-dependent.

**Funding:** This work was supported by Japan Society for the Promotion of Science (JSPS), https://www.jsps.go.jp/english/index.html, KAKENHI Grant Numbers 17H04167 (A.H.), 18K07920 (J.I.), and 21K11603 (T.H.), and by Japan Agency for Medical Research and Development (AMED), https://www.amed.go.jp/en/index.html, under Grant Numbers JP21ek0109416 (N.N.), JP22fk0210073 (T.I.), JP23fk0210096 (N. N.). The funders had no role in study design, data collection and analysis, decision to publish, or preparation of the manuscript.

**Competing interests:** The authors have declared that no competing interests exist.

## Introduction

Ursodeoxycholic acid (UDCA) is synthesized in the liver and is a major primary bile acid (BA) in nutria [1] and some species in Ursidae [2]. In humans, UDCA is not a primary BA and is made by intestinal bacteria from a primary BA, chenodeoxycholic acid (CDCA) [3]. Therefore, UDCA is not a principal constituent of the total BA pool in humans. More than a half-century ago, UDCA was reported as a unique BA having cytoprotective and anti-cholestatic properties. Since then, UDCA has been used for the treatment of various hepatobiliary diseases, including cholelithiasis, primary biliary cholangitis, chronic hepatitis, and drug-induced liver injury [4].

The mouse is the most commonly used laboratory animal to extrapolate investigations about human hepatobiliary diseases such as cholelithiasis [5], autoimmune hepatitis [6], primary biliary cholangitis [6], primary sclerosing cholangitis [6], and nonalcoholic steatohepatitis [7]. However, the BA compositions in mice and humans are substantially different. For example, cholic acid (CA) and CDCA are end products in the human liver, whereas CDCA is further metabolized to muricholic acids (MCAs) by CYP2C70 in the mouse liver (Fig 1) [8]. In addition, CA and CDCA are metabolized by the gut microbiota into secondary BAs, deoxycholic acid (DCA) and lithocholic acid (LCA), respectively, and mouse liver CYP2A12 but not human liver enzymes can convert these secondary BAs to primary BAs [8]. Therefore, the hepatobiliary concentrations of secondary BAs are almost deficient in mice compared with those in humans.

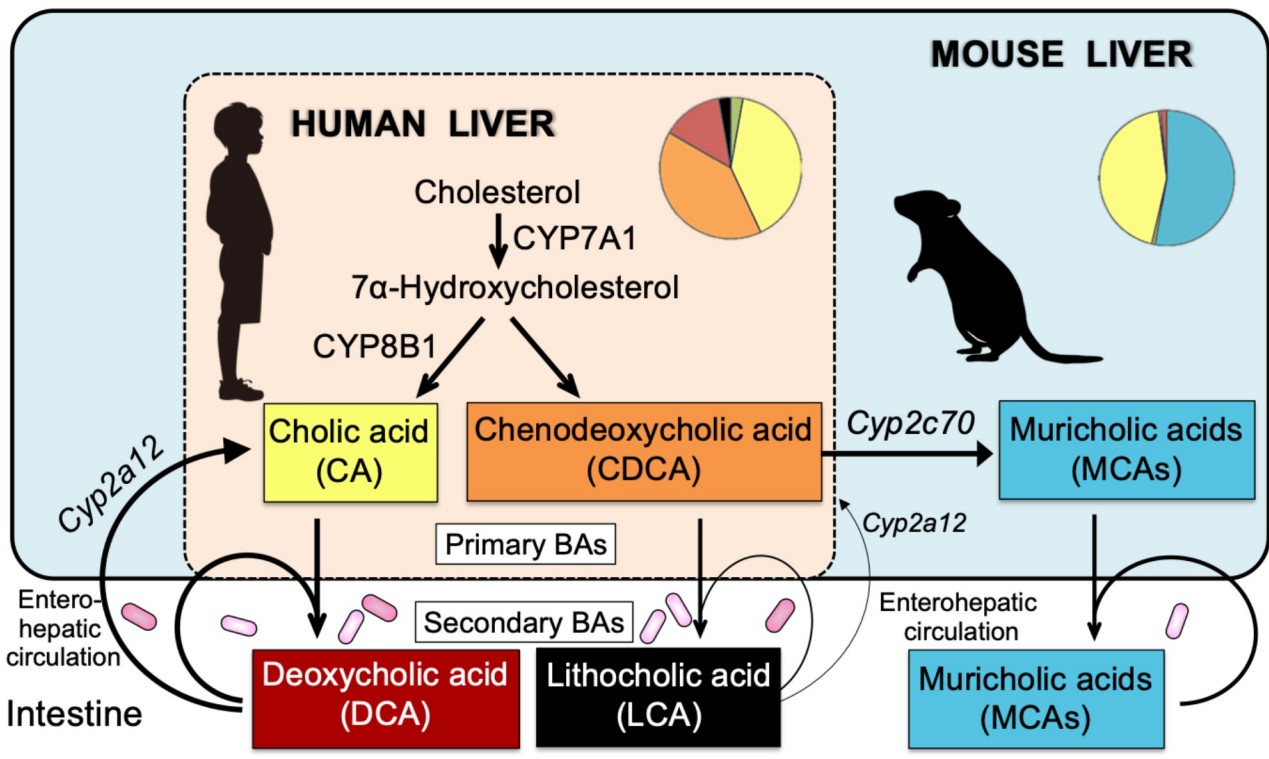

**Fig 1. Comparison of bile acid (BA) metabolism between human and mouse.** CA and CDCA are end products in the human liver, but the mouse liver further metabolizes CDCA to MCAs by CYP2C70. In addition, the mouse liver can convert secondary BAs, DCA and LCA, into CA and CDCA by CYP2A12. Because of the low intestinal re-absorption rate (human and mouse) and small CDCA pool (wild-type mouse), the enterohepatic circulation of LCA is usually less (thin arrows) than those of other BAs (thick arrows). Knockout of the *Cyp2a12* and *Cyp2c70* genes results in mice with a human-like hydrophobic BA composition. *Cyp2a12* and *Cyp2c70* are genes responsible for hepatic BA 7α-hydroxylation and CDCA 6β-hydroxylation, respectively.

To overcome these limitations, we recently generated *Cyp2a12*[-/-]*Cyp2c70*[-/-] double knock-out (DKO) mice. These mice have a human-like BA composition lacking MCAs and increasing CDCAs, DCAs, and LCAs. However, due to the increased hydrophobicity of BAs, these mice exhibited liver injury, reduced fertility, and high post-weaning mortality [8]. UDCA is less hydrophilic than mouse MCAs but more hydrophilic than human principal BAs, CA, CDCA, DCA, and LCA [9]. Therefore, we hypothesized that UDCA administration might improve liver damage, fertility, and post-weaning mortality in DKO mice.

In the present study, we orally administered UDCA to both male and female DKO mice and evaluated the effects on the health problems of these mice. The results demonstrated that UDCA significantly improved fertility and post-weaning mortality in DKO mice. UDCA also improved liver injury in DKO mice at six weeks of age in both sexes. However, at 20 weeks of age, the improvement in liver damage was maintained only in males.

## Materials and methods

### Animals

*Cyp2a12*[-/-]*Cyp2c70*[-/-] DKO mice were prepared in the manner described previously [8]. Mice were kept under a regular 12-hour light-dark cycle (light period: 6:00–18:00) with free access to regular chow (CRF-1, Oriental Yeast Co., Ltd., Japan) and water in pathogen-free conditions. The UDCA solution was prepared by diluting ursodeoxycholate 2367.4 mg/100 mL injection (DS Pharma Animal Health Co., Ltd., Osaka, Japan) with distilled water to a final concentration of 75 mg/100 mL. This solution was administered in the drinking water after weaning at three weeks of age, equivalent to a dose of 150 mg/kg/day. Matings were made at nine weeks of age (n = 85). Pregnancy and delivery rates were compared between groups treated with UDCA (n = 75) and untreated controls (n = 10). The litters were also maintained with UDCA and divided randomly into two groups: one (male, n = 16; female, n = 13) in which UDCA was discontinued at six weeks of age and the other (male, n = 4; female, n = 4) was continued until 20 weeks of age. After fasting for four hours with free access to water, the animals were euthanized by exsanguination under combination anesthesia with medetomidine, midazolam, and butorphanol. The serum, gallbladder, liver, small intestine, and feces were collected and frozen at –80°C until analysis. The Animal Experiment Committee of Tokyo Medical University approved the experimental design (Permission #H30-0069, H31-0064, R2-0013, and R3-0010).

### Liver function analysis

Serum activities of alanine and aspartate transaminases (ALT and AST) and alkaline phosphatase (ALP) were measured by colorimetric assays using Transaminase CII-Test Wako and LabAssay ALP (FUJIFILM Wako Pure Chemical Corporation, Osaka, Japan), respectively.

### Determination of BA concentrations

The concentrations of each BA in the liver, bile, small intestine, feces, and serum were measured using a LC-MS/MS method, as described previously [8].

### Determination of lipid concentrations

Total cholesterol concentrations in the liver and serum were determined by colorimetric assay using Cholesterol E-Test Wako (FUJIFILM Wako Pure Chemical Corporation). In addition, serum and hepatic concentrations of sterols and oxysterols were determined using the previously described LC-MS/MS method [8].

## Histopathological examination

Liver samples were fixed in 10% neutral buffered formalin and embedded in paraffin blocks. Each paraffin block was cut into 4 μm sections and stained using hematoxylin and eosin. Immunohistochemical (IHC) stain was carried out using an auto-stain system with the exclusive reagents (Discovery XT system, VENTANA®, Roche Diagnostics K.K., Basel, Schweiz) and Mouse on Mouse blocking method [10]. Deparaffinized 4 μm thick specimens of liver tissue were heated with Tris-EDTA buffer pH 7.8 or citrate-based buffer pH 6.5 at 95°C for 30–60 minutes for antigen retrieval and then were blocked with purified normal mouse IgG (FUJIFILM Wako Pure Chemical Corporation) and Protein Concentrate solutions (M.O.M.® Immunodetection Kit, Vector Laboratories, Burlingame, CA) for 1 hour and 5 minutes, respectively. After that, the specimens were incubated with each of the primary polyclonal antibodies of F4/80 (1:400; Proteintech®, Rosemont, IL), cytokeratin 19 (CK19, 1:4000; Proteintech®), and myeloperoxidase (MPO, pre-diluted; VENTANA®) for 32 min at 37°C. Thereafter, the incubations with secondary antibody and detection were developed using the VENTANA ultraView universal DAB detection kit (VENTANA®), and the nucleus and cytoplasm were stained by the Hematoxylin II Counterstain (VENTANA®) and the bluing reagent (VENTANA®), respectively. Specific immunoreaction of the primary antibodies was confirmed by incubation without each antibody.

## Determination of mRNA expression

An aliquot of the liver specimen was stored in RNAlater (Thermo Fisher Scientific) at -80°C until RNA isolation. Then, as described previously [8], total RNA extraction, reverse transcription, and real-time quantitative PCR were performed. The sequences of the oligonucleotide primer pairs used to amplify mRNAs are shown in the S1 Table. mRNA expression levels were standardized to those of *Gapdh*, and the mean expression levels in male UDCA-untreated mice were set to 1.0.

## Preparation of liver microsomes

The liver specimen was homogenized with a loose-fitting Teflon pestle in 4 volumes of 3 mM Tris-HCl buffer (pH 7.4) containing 0.25 M sucrose and 0.1 mM EDTA. The homogenate was centrifuged at 700 g for 10 min, the supernatant was re-centrifuged at 7,000 g for 20 min, and the supernatant was further centrifuged at 105,000 g for 90 min. The pellet was suspended in 100 mM potassium phosphate buffer (pH 7.4) containing 1 mM EDTA, 5 mM dithiothreitol, 50 mM KCl, and 20% glycerol (v/v).

## Enzyme assays

The activities of hepatic microsomal 6α-hydroxylation, 6β-hydroxylation, and 7α-hydroxylation of taurolithocholic acid (TLCA) were measured according to a method described previously [8]. Briefly, microsomes were incubated with 200 μM of TLCA and NADPH generating system, and BAs were extracted with Bond Elut C18 cartridges. Taurohyodeoxycholic acid (THDCA), tauromurideoxycholic acid (TMDCA), and TCDCA were quantified by LC-MS/MS. In addition, hepatic microsomal cholesterol 7α-hydroxylase (CYP7A1) activities were assayed as described previously [11].

## Determination of serum FGF15 concentrations

Serum fibroblast growth factor 15 (FGF15) levels were measured by mouse FGF15 ELISA kit (Catalog #MBS2700661, MyBiosource, Inc., San Diego, CA), according to the manufacturer's instructions.

## Statistics

Data are expressed as the mean ± standard error of the mean (SEM). Statistical significance of the contingency between the two kinds of classification was evaluated using the Fisher's exact test. Statistically significant differences between groups were evaluated using one-way ANOVA and post-hoc Tukey–Kramer test. For all analyses, significance was accepted at the level of $p < 0.05$. Correlations were tested by calculating non-parametric Spearman's rank correlation coefficient, $r_s$. All statistical analyses were conducted using Prism (ver. 9.1.0) software (GraphPad Software, San Diego, CA, USA).

## Results

### Effects of UDCA administration on the breeding efficiency of DKO mice

The reproductive capacity of DKO mice was lower than that of wild-type mice, and 20% to 25% of the mice unexpectedly died within a week of weaning (four weeks of age) [8]. UDCA administration did not improve the pregnancy rate or the number of pups per litter but significantly improved the normal delivery rate by inhibiting miscarriage and preterm birth (Table 1). In addition, UDCA treatment completely prevented unexpected deaths after weaning.

### Effects of UDCA treatment on liver injury of DKO mice

Liver function tests at 20 weeks of age were compared between DKO mice continuously treated with UDCA and those discontinued UDCA at six weeks of age (Fig 2A). In UDCA-untreated (discontinued at six weeks) 20-week-old mice, serum ALT activities were above the normal range. UDCA treatment significantly reduced the ALT levels in male but not female mice. Serum AST activities were normal or marginally elevated in UDCA-untreated mice, and UDCA treatment tended to increase the levels in female mice, although the difference was not statistically significant. ALP activities in both sexes were almost normal and were not changed significantly by UDCA administration. We compared liver histopathology among UDCA-untreated and treated 20-week-old mice in both sexes (Fig 2B). In comparison, UDCA-treated six-week-old DKO mice showed no significant histological findings (S1 Fig), supported by normal serum ALT activities (S2 Fig). In both sexes of 20-week-old mice, we observed bile ductular reaction, infiltration of neutrophils and lymphocytes, and focal necrosis of

**Table 1. Effects of UDCA administration on the fertility of DKO mice.**

| Parameter | Groups | With UDCA | Without UDCA | P-value |
|---|---|---|---|---|
| **Pregnancy / Mating** | Mating | n = 75 | n = 10 | NS |
| | Pregnancy | 53 [a] (70.7%) | 8 (80.0%) | |
| | No pregnancy | 22 (29.3%) | 2 (20.0%) | |
| **Delivery / Pregnancy** | Pregnancy | n = 43[a] | n = 8 | < 0.005 |
| | Normal delivery | 43 (100%) | 5 (62.5%) | |
| | Abnormal delivery[b] | 0 (0%) | 3 (37.5%) | |
| **Number of pups / Litter** | Normal delivery | n = 43 | n = 5 | — |
| | Number of pups | 226 | 26 | |
| | Per litter | 5.3 | 5.2 | |

[a]Ten pregnant individuals were euthanized for population control.

[b]Abnormal delivery includes miscarriage and preterm birth. Fisher exact tests were used to assess the association of UDCA administration with pregnancy and delivery rates. NS, not significant.

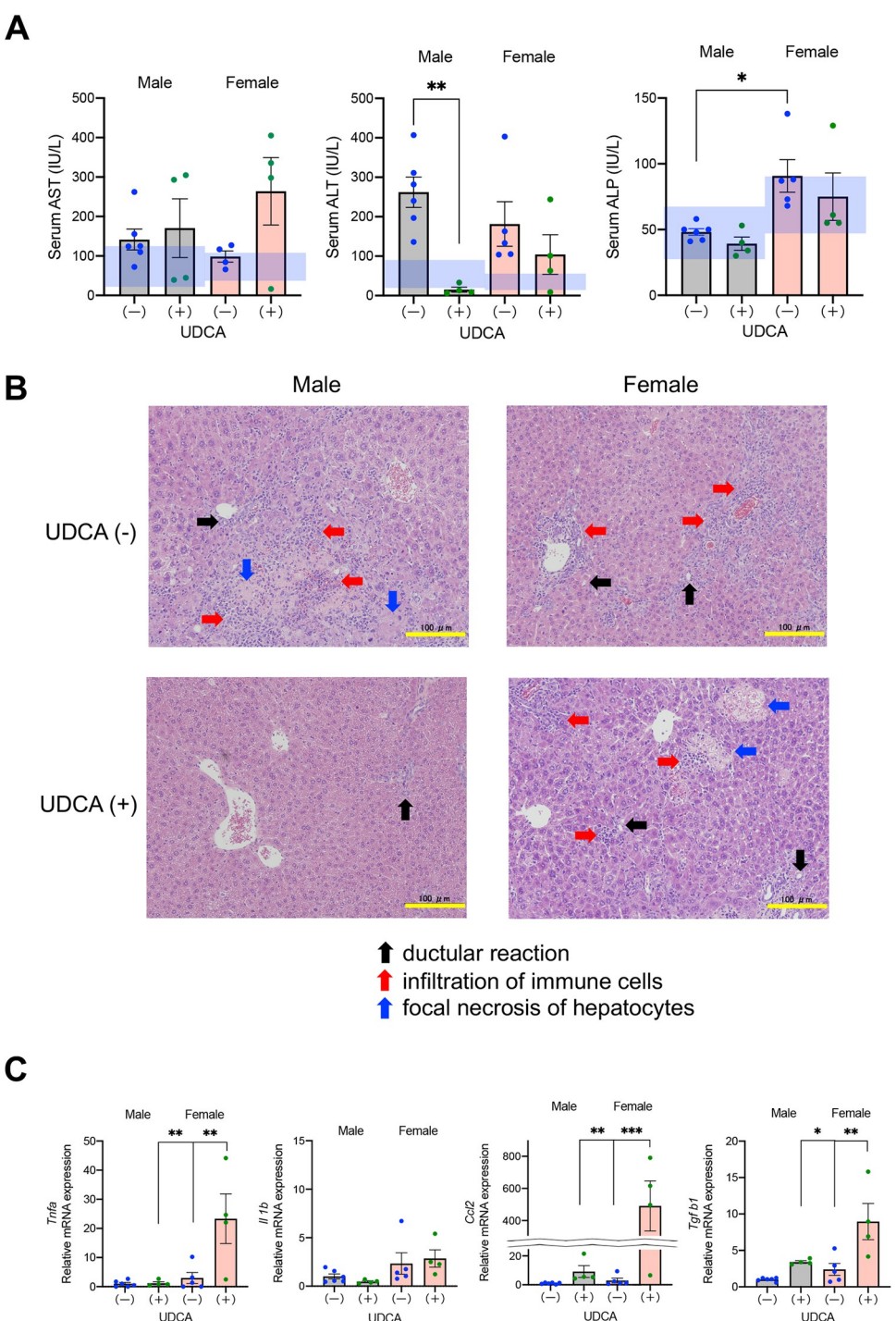

**Fig 2. Effects of UDCA treatment on liver injury.** DKO mice were bred under UDCA administration, and offspring in both sexes were divided into two groups: one stopped receiving UDCA at six weeks of age while the other continued UDCA until 20 weeks. (A) Serum activities of AST, ALT, and ALP in mice at 20 weeks with (+) or without (discontinued) (-) UDCA treatment. The blue shaded areas indicate the normal ranges. (B) Representative histopathologic features of the livers at the age of 20 weeks with or without UDCA treatment. Hematoxylin/eosin stain. Scale bars, 100 μm. Arrows indicate ductular reaction (black), infiltration of immune cells (red), and focal necrosis of hepatocytes (blue). (C) Hepatic mRNA expression levels of inflammatory cytokines in mice at 20 weeks with or without UDCA treatment. Each column and error bar represents the mean and SEM. *P < 0.05, **P < 0.01, and ***P < 0.001 were considered statistically significantly different by the Tukey-Kramer test.

hepatocytes, irrespective of UDCA treatment. However, the chronic inflammation and hepatocyte damage were generally mild in male UDCA-treated mice compared with the other three groups of mice. The proliferation of bile ductular structure was confirmed by IHC stain of CK19 (S3 Fig). Infiltration of macrophages were assessed by IHC stain of F4/80 and mRNA expression levels of *Cd68* and *Cd163* (S4 Fig), and that of neutrophils and T cells were determined by IHC stain of MPO and mRNA expression levels of *Mpo*, *Cd4*, and *Cd8* (S5 Fig).

Hepatic mRNA expression levels of inflammatory cytokines, such as tumor necrosis factor-α (*Tnfa*), interleukin-1β (*Il1b*), chemokine (C-C motif) ligand 2 (*Ccl2*), and transforming growth factor-β1 (*Tgfb1*) were compared in 20-week-old mice treated with and without (discontinued) UDCA. UDCA treatment did not affect *Il1b* expression but significantly upregulated *Tnfa*, *Ccl2*, and *Tgfb1* in female mice (Fig 2C).

## Effects of UDCA treatment on BA pool and fecal BA excretion

We compared the composition of BAs in the enterohepatic circulation and feces in 20-week-old DKO mice treated with or without UDCA. The total BA pool in each group was calculated by summing the amounts of all BAs in the liver, gallbladder, and small intestine. Although glycine-conjugated BAs were also detected, they were excluded because their proportion was less than 1% of taurine-conjugated BAs. UDCA administration markedly increased proportions and amounts of UDCAs and decreased CDCAs in the total BA pool (Fig 3A, S2 and S3 Tables). The total BA pool in female DKO mice was significantly larger than in males, irrespective of UDCA treatment, and the pools tended to increase after treatment with UDCA in both sexes (Fig 3B). In addition, UDCA treatment markedly increased fecal excretion of BAs (Fig 3B). In UDCA-treated DKO mice, the fecal proportions of UDCAs were significantly increased, but the most abundant fecal BAs in these mice were unconjugated LCA in both sexes (Fig 3A, S4 and S5 Tables).

## Effects of UDCA treatment on BA composition and concentrations in serum, liver, and bile

To investigate sex difference mechanisms of UDCA induced liver injury, we compared BA composition, concentrations, and hydrophobicity indexes in serum, liver, and bile among male and female DKO mice with or without UDCA treatment. The analyses of BA compositions and concentrations in the liver indicated that the proportion (Fig 4A and S6 Table) and concentration (Fig 4B and S7 Table) of TLCAs were markedly elevated in female UDCA-treated mice compared with male UDCA-treated mice. However, biliary TLCA proportions were quite similar between male and female UDCA-treated mice (Fig 4A and S8 Table), although the total biliary BA amount in female mice was more than double that in male mice (Fig 4B and S9 Table). Regarding serum BA compositions, conjugated BAs were increased in UDCA-treated mice compared with UDCA-untreated mice (Fig 4A and S10 Table). The proportions (Fig 4A and S10 Table) and concentrations (Fig 4B and S11 Table) of serum LCAs were not significantly different between male and female mice treated with UDCA. UDCA treatment significantly decreased serum and biliary hydrophobicity indexes in both sexes of mice (Fig 4C). The liver hydrophobicity index was also decreased in male UDCA-treated mice but was unchanged in female UDCA-treated mice due to the accumulation of TLCA. Serum ALT activities were highly correlated with hydrophobicity indexes in the liver ($r_s = 0.9127$, n = 19, p < 0.0001), suggesting the accumulation of TLCA in the liver causes liver injury in female mice after UDCA treatment.

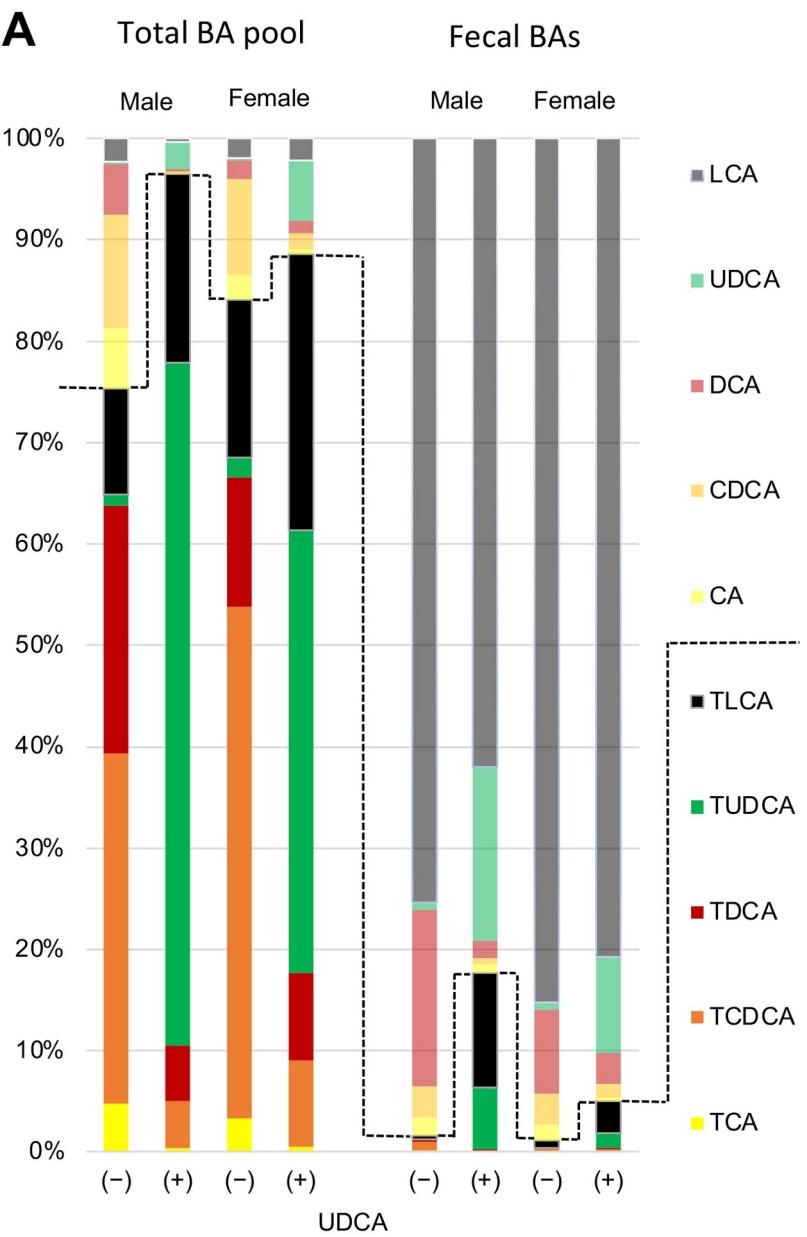

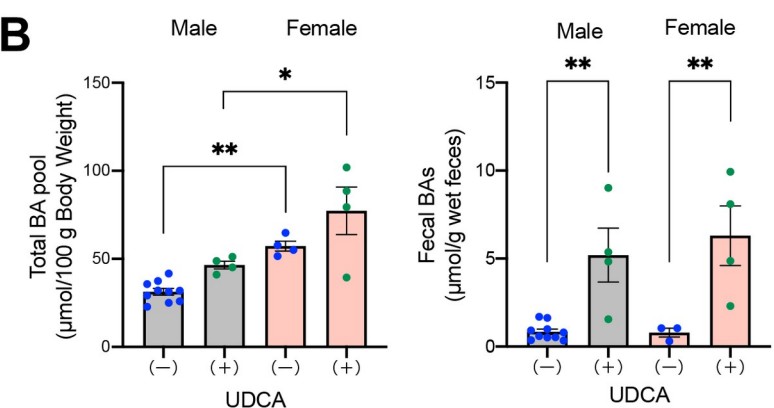

**Fig 3. Effects of UDCA treatment on BA pool and fecal BA excretion.** DKO mice at 20 weeks of age, with (+) or without (-) UDCA treatment, were compared. (A) Composition of BAs in the total BA pool and feces. The total BA pool in the enterohepatic circulation was calculated by adding total BAs in the liver, gallbladder, and small intestine. The broken line indicates the boundary between conjugated and unconjugated BAs. (B) BA amounts in the total pool and feces. Each column and error bar represents the mean and SEM. *P < 0.05 and **P < 0.01 were considered statistically significantly different by the Tukey-Kramer test.

## Comparative activities of LCA detoxification between male and female DKO mice

To explore why UDCA treatment causes hepatic TLCA accumulation and liver injury, especially in female DKO mice, we measured LCA detoxification activity in the liver. As shown in Fig 5A, most LCA is conjugated with taurine in the liver. Although WT mice convert almost all hepatic TLCA to T-βMCA through TCDCA; DKO mice do not have this pathway. Therefore, three additional pathways detoxifying LCAs by 3α-sulfation, 6α-hydroxylation, and 6β-hydroxylation [12] appear important in DKO mice. SULT2A1 catalyzes 3α-sulfation of LCA

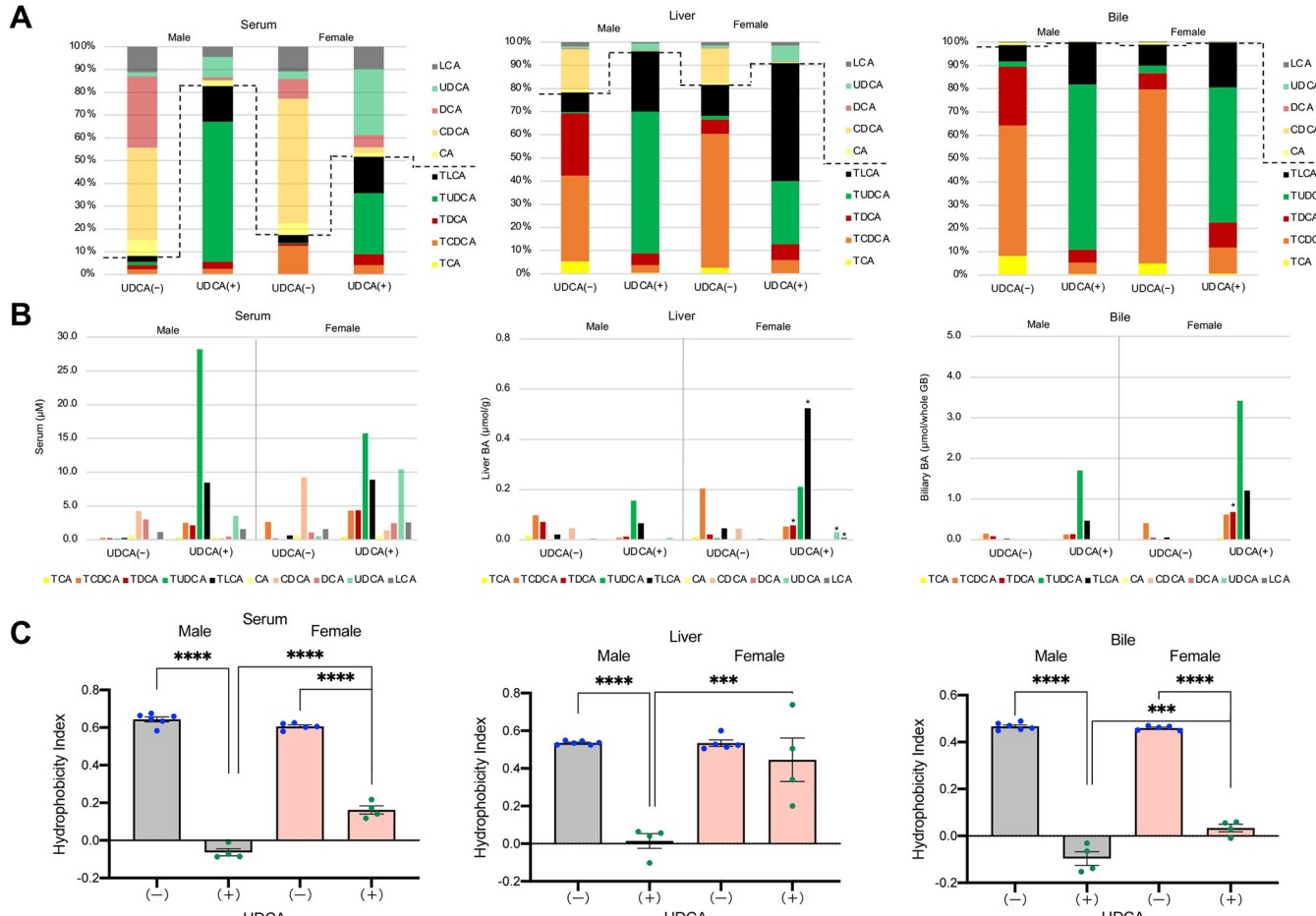

**Fig 4. Comparison of BA compositions, concentrations, and hydrophobicity indices among serum, liver, and bile.** DKO mice at 20 weeks of age, with (+) or without (-) UDCA treatment, were compared. (A) BA compositions in serum, liver, and bile. The broken lines indicate the boundary between conjugated and unconjugated BAs. (B) BA concentrations in serum, liver, and bile. *P < 0.05 was considered statistically significantly different from UDCA treated male mice by the Tukey-Kramer test. (C) Hydrophobicity indices of total BAs in serum, liver, and bile. Each column and error bar represents the mean and SEM. **P < 0.01, ***P < 0.001, and ****P < 0.0001 were considered statistically significantly different by the Tukey-Kramer test.

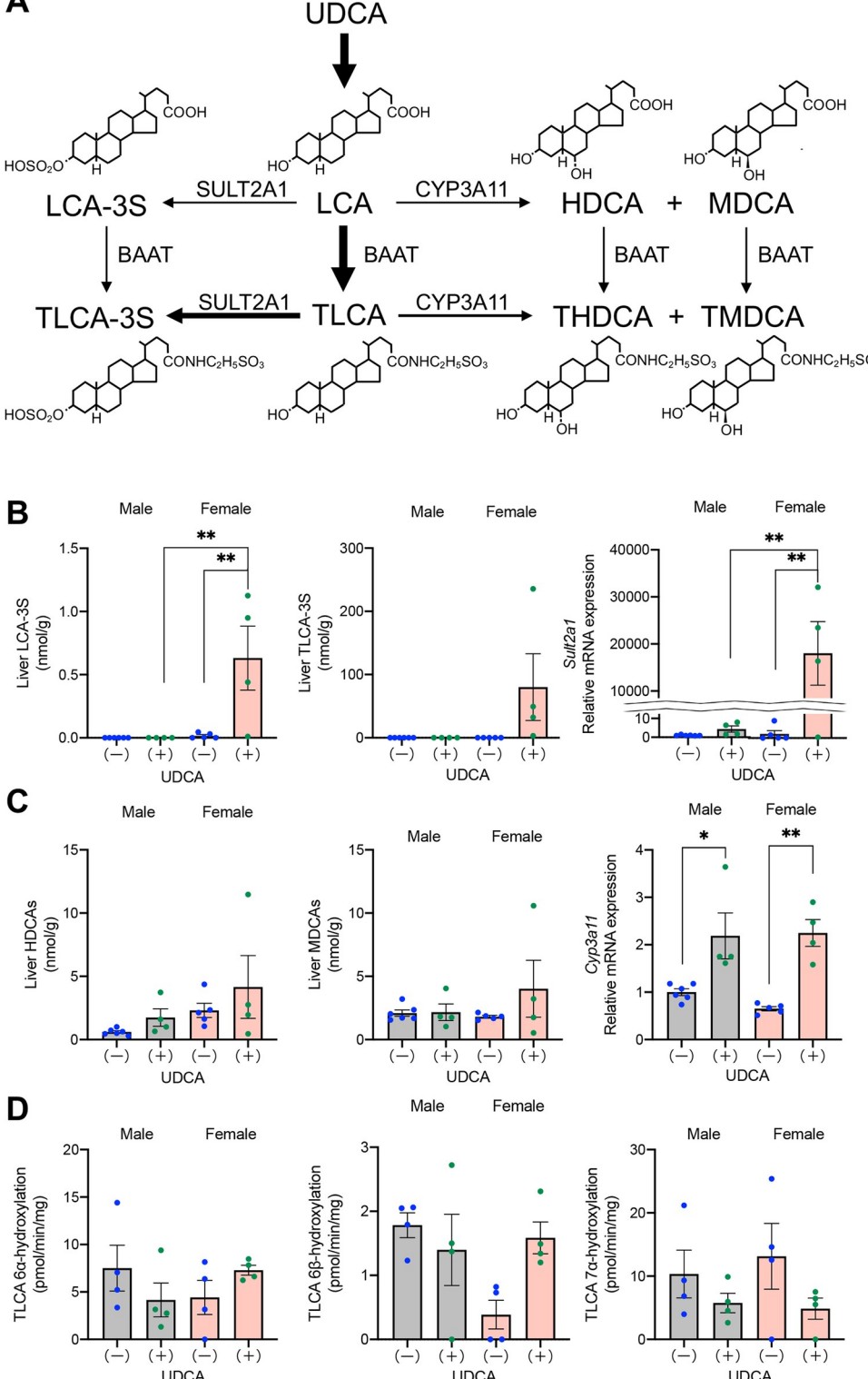

**Fig 5. Comparison of LCA detoxification activities.** DKO mice at 20 weeks of age, with (+) or without (-) UDCA treatment, were compared. (A) Proposed metabolic pathways for LCA in mice. (B) Hepatic concentrations of LCA-3S and TLCA-3S and mRNA expression levels of *Sult2a1*. (C) Hepatic concentrations of HDCAs (HDCA + THDCA) and MDCAs (MDCA + TMDCA) and mRNA expression levels of *Cyp3a11*. (D) Hepatic microsomal enzyme activities that catalyze 6α-, 6β-, and 7α-hydroxylations of TLCA. Each column and error bar represents the mean and SEM. *P < 0.05 and **P < 0.01 were considered statistically significantly different by the Tukey-Kramer test.

and TLCA to form LCA-3-sulfate (LCA-3S) and TLCA-3S. However, hepatic LCA-3S and TLCA-3S concentrations and mRNA expression levels of *Sult2a1* did not support the hypothesis that SULT2A1 was downregulated in UDCA-treated female mice (Fig 5B). Hydroxylation of LCA (TLCA) at 6α- and 6β-positions are catalyzed by CYP3A11 and form HDCA (THDCA) and MDCA (TMDCA), respectively. However, the possibility that CYP3A11 was downregulated in UDCA-treated female mice was also ruled out by the results of HDCA and MDCA concentrations and mRNA expression levels of *Cyp3a11* in the liver (Fig 5C).

In addition to determining hepatic BA concentrations and mRNA expression levels, we directly measured hepatic microsomal enzyme activities that catalyze 6α-, 6β-, and 7α-hydroxylation of TLCA (Fig 5D). However, a significant reduction of these enzyme activities was not observed in UDCA-treated female mice. In DKO mice, we detected significant activity of TLCA 7α-hydroxylation, despite lacking *Cyp2a12*. Therefore, it is suggested that TLCA 7α-hydroxylase activity is catalyzed not only by CYP2A12 but also by CYP2A22 [8]. However, the fact that TLCA 7α-hydroxylase was not upregulated in UDCA-treated DKO mice indicates that the accumulated LCAs do not induce *Cyp2a22*.

## Regulation of Cyp3a11 and Sult2a1 expression by nuclear receptors in DKO mice

As shown in Fig 5C and 5B, *Cyp3a11* in both sexes and *Sult2a1* in females were significantly upregulated by treatment with UDCA. Nuclear receptors, pregnane X receptor (PXR, NR1I2), constitutive androstane receptor (CAR, NR1I3), and vitamin D receptor (VDR, NR1I1), are all known to stimulate the expression of *Cyp3a11*. However, *Cyp2b10*, a target gene of CAR, was not upregulated in UDCA-treated mice (Fig 6A). Therefore, PXR and VDR appear to be primary LCA receptors that upregulate hepatic *Cyp3a11* in this model. Conversely, the regulation of *Sult2a1* is complex. Previous studies have shown that CAR, VDR, hepatocyte nuclear factor 4α (HNF4α, NR2A1), and liver X receptor α (LXRα, NR1H3) upregulate while PXR and FXR downregulate *Sult2a1*. In our UDCA treated mice, the stimulation of *Sult2a1* was not explained by the activation of CAR (target gene: *Cyp2b10*) or HNF4α or deactivation of PXR (target gene: *Cyp3a11*) or FXR (target genes: *Shp and Bsep*). Because LXRα target genes, i.e., *Srebp1*, *Abcg5*, and *Abcg8*, were virtually upregulated in these mice, the activation of LXRα and VDR may have stimulated *Sult2a1* in this model.

To determine the reason for the activation of LXRα in UDCA-treated DKO mice, we quantified hepatic concentrations of cholesterol and LXRα ligand oxysterols. As shown in Fig 6B, both sexes of UDCA-treated mice showed significantly elevated hepatic cholesterol and 25-hydroxy-cholesterol concentrations. Hepatic cholesterol accumulation was not due to increased cholesterol biosynthesis, intestinal cholesterol absorption, or decreased biliary cholesterol excretion. Hepatic expression levels of HMG-CoA reductase, the rate-limiting enzyme in the cholesterol biosynthetic pathway, and serum lathosterol and 7-dehydrocholesterol concentrations, indicators of cholesterol biosynthesis, were not increased in UDCA-treated DKO mice (Fig 6C). In addition, serum concentrations of sitosterol and campesterol, surrogate markers of cholesterol absorption, did not increase (Fig 6C), and hepatic *Abcg5* and *Abcg8*, transporters for the excretion of cholesterol into bile, did not decrease in these mice (Fig 6A). Since UDCA-treated mice tended to have downregulated *Cyp7a1* and *Cyp8b1*, the inhibited classic bile acid biosynthetic pathway may explain the increased hepatic cholesterol concentrations in these mice (Fig 6D).

## Discussion

In DKO mice without UDCA treatment, the infertility rate was approximately 20% (Table 1), higher than that of wild-type C57BL/6J mice (~6%, unpublished observation). Testes express

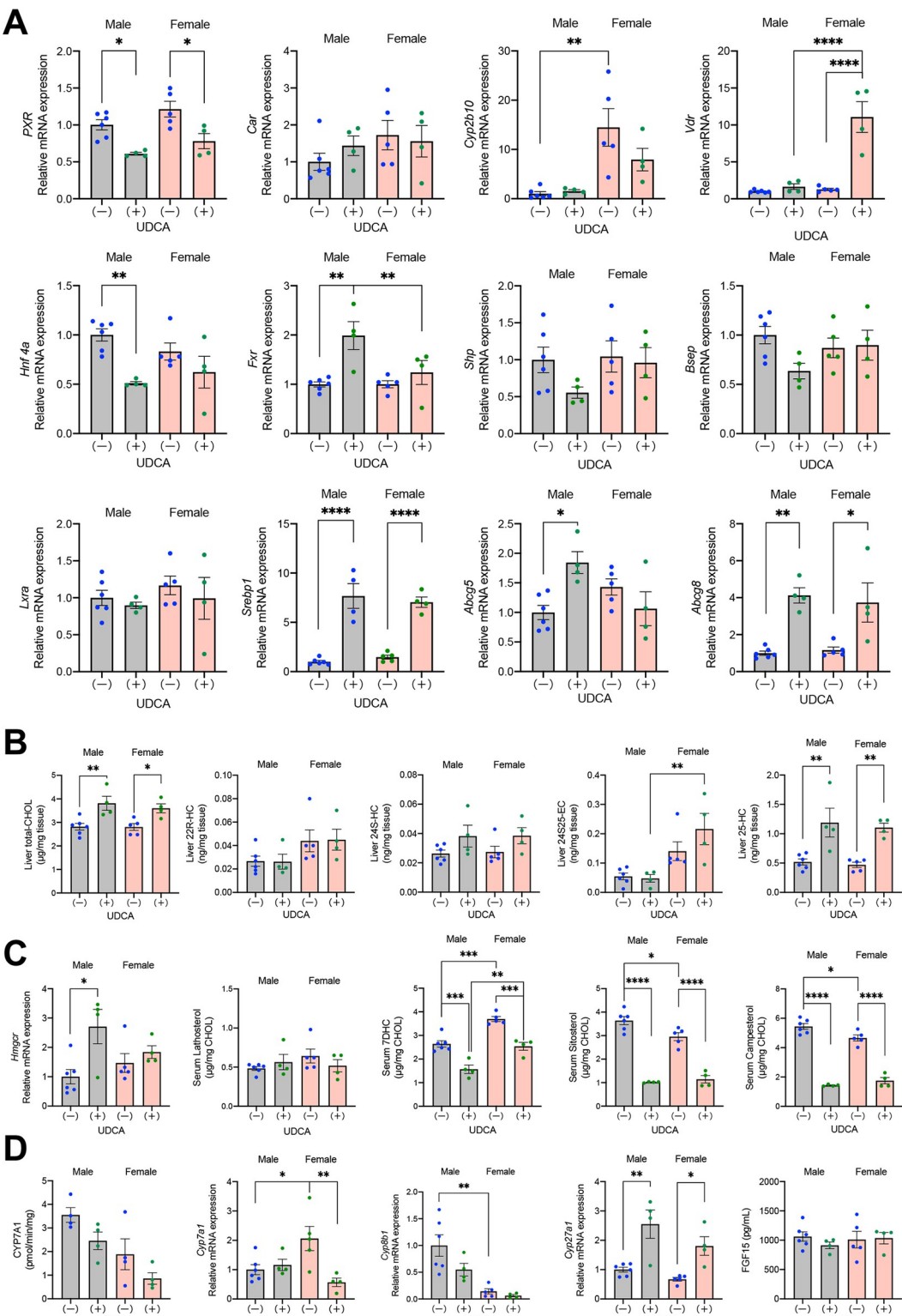

**Fig 6. Effects of UDCA treatment on hepatic cholesterol and BA metabolism.** DKO mice at 20 weeks of age, with (+) or without (-) UDCA treatment were compared. (A) Hepatic mRNA expression levels of nuclear receptors and their target genes. (B) Hepatic concentrations of cholesterol and oxysterols. CHOL, cholesterol; HC, hydroxycholesterol; EC, epoxycholesterol. (C) Hepatic mRNA expression levels of the key enzyme in the cholesterol biosynthetic pathway and serum sterol concentrations associated with cholesterol metabolism. (D) Hepatic activities and mRNA expression levels of key

enzymes in the BA biosynthetic pathways and serum concentrations of FGF15. Each column and error bar represents the mean and SEM. *P < 0.05, **P < 0.01, ***P < 0.001, and ****P < 0.0001 were considered statistically significantly different by the Tukey-Kramer test.

BA responsive membrane receptor, Takeda G-protein receptor 5 (TGR5) [13], and nuclear FXR [14]. It has been reported that BAs reduce sperm production through the activation of TGR5 [15]. In addition, the activation of FXR induces apoptosis of germ cells by decreasing testosterone synthesis [16] or suppresses sperm production by maintaining an undifferentiated germ cell pool in an androgen-independent manner [17]. In females, the roles of BAs on oocyte maturation are still unknown, however effects are suspected [18]. Though, the deactivation of TGR5 and FXR by UDCA treatment did not improve the low pregnancy rate of the DKO mice.

On the other hand, UDCA treatment markedly improved frequent miscarriage and preterm birth (Table 1) and unexpected deaths after weaning [8]. Frequent preterm birth is a characteristic feature of intrahepatic cholestasis of pregnancy (ICP) [19]. A recent report demonstrated that serum total BA concentrations directly correlated with the preterm birth rates in pregnant women, including ICP [20]. In addition, serum BAs but not liver injury induced preterm delivery in pregnant mice [20]. The upregulation of myometrial oxytocin receptors by CA is thought to be one of the factors for preterm delivery [21]. In female DKO mice, serum total BA concentration was more than twice as high as that in female wild-type mice [8]. UDCA treatment further increased the serum total BA concentration in female DKO mice but completely prevented abnormal delivery. Although no reports compare the effects of different BAs on preterm birth, reducing the serum BA hydrophobicity index by UDCA treatment might have inhibited abnormal delivery (Fig 4B and 4C). In contrast, UDCA treatment was ineffective in reducing adverse perinatal outcomes in patients with ICP [22]. Therefore, it is unclear whether pregnant DKO mice would be a good model for ICP.

Discontinuation of UDCA at six weeks of age returned the BA composition to the original human-like composition and induced liver injury within two weeks (S2 Fig). On the other hand, continuous administration of UDCA prevented hepatotoxicity in males, but liver damage appeared in females after raising until 20 weeks of age (Fig 2B). Therefore, we performed a BA analysis of each organ to determine the reason for liver damage in UDCA-treated female DKO mice. BA composition in the enterohepatic circulation (total pool) and feces indicated that most UDCA was transformed into LCA by gut microbiota, and higher proportions of TLCA in the enterohepatic circulation were observed in DKO mice with UDCA than in those without UDCA (Fig 3A). In addition, the proportion and concentration of LCAs in the liver from UDCA-treated mice were significantly higher in females than in males. However, serum and biliary proportion and concentration of LCAs were not significantly different between UDCA-treated male and female mice (Fig 4A and 4B). These results suggest that 20-week-old female DKO mice are more likely to accumulate TLCA in the liver when UDCA is administered. As a result, UDCA treatment did not reduce the BA hydrophobicity index, only in the liver of female mice (Fig 4C). In addition, serum ALT activities varied among UDCA-treated female mice, and the mouse with higher ALT levels had a higher concentration of TLCA in liver tissue.

Why does TLCA accumulate in the liver of female DKO mice after long-term administration of UDCA? LCAs are sulfate-conjugated at the 3α-position in the liver to form LCAs-3S. This conjugation is a significant pathway to protect against LCA-induced hepatotoxicity [23], and SULT2A1 catalyzes this reaction [24]. However, our data suggest that reducing SULT2A1 activity does not cause the accumulation of TLCA in the UDCA-treated female mice because hepatic *Sult2a1* expression and the concentrations of hepatic LCAs-3S increased in the female

mice (Fig 5B). Other pathways to detoxify LCAs are hydroxylations at the 6α and 6β positions to form HDCAs and MDCAs, respectively. Human CYP3A is known to catalyze both reactions [25], and the latter reaction may also be catalyzed by CYP2C70 in WT mice [26] but not in DKO mice. However, hepatic *Cyp3a11* expression, activity, and concentrations of hepatic HDCA and MDCA did not decrease in UDCA-treated female mice (Fig 5C and 5D). Thus, we cannot elucidate the cause of TLCA accumulation in the liver of female mice by reducing detoxifying enzyme activity.

Another possible cause of TLCA accumulation in the liver of female mice is the insufficient biliary excretion of TLCA due to higher production of LCAs and/or larger BA pool size, including LCAs. For the former, the expression of hepatic *Cyp8b1* was significantly lower in females than in males (Fig 6D), which suggests that the female CDCA/CA production ratio is higher than that in males. In addition to the exogenously added UDCA, significant CDCA synthesis may have contributed to the higher production of LCA in female DKO mice. For the latter, in both wild-type C57BL/6 and CD-1 mice, the BA pools are larger in females than males [27], which may be the opposite of what has been found in humans [28]. In addition, liver BA concentrations, including TLCA, were also higher in female C57BL/6 mice than males [29]. In our study, the BA pool in female DKO mice was larger than in males irrespective of UDCA treatment (Fig 3B). It is not clear whether TLCA is an excellent substrate for mouse BSEP like human BSEP [30], but the hepatic expression of BSEP did not upregulate despite the accumulated LCA in UDCA-treated female mice (Fig 6A). LCA is reported to decrease expression of BSEP through FXR antagonist activity [31], which may explain the reason for the inhibited upregulation of BSEP in the female DKO mice. It may be mentioned here that basolateral BA efflux transporters are also important for hepatic BA clearance. Organic solute transporter β (OSTβ) is regulated by FXR, while multidrug resistance-associated proteins 3 and 4 (MRP3 and MRP4) are controlled by FXR independent manner [32]. In UDCA-treated female DKO mice, these transporters were not significantly upregulated either (S6 Fig).

The increased serum LCA concentration from high-dose UDCA treatment has also been reported in patients with primary sclerosing cholangitis [33]. These patients received UDCA at a dose of 28 to 30 mg/kg/day, while we calculated the UDCA amount administered to our DKO mice was approximately 150 mg/kg/day. The patients' proportions of serum LCA, DCA, UDCA, CDCA, and CA were 8%, 8%, 74%, 4%, and 6%, respectively, which were similar to DKO mice treated with UDCA. In addition, female and older patients were reported to be more likely to have a marked increase in their LCA concentration after UDCA treatment [33]. Thus, as far as the serum was compared, BA composition after high-dose UDCA administration was similar in humans and DKO mice. Our aged female mice showed an unexpected increase in hepatic TLCA proportion compared with serum and bile. Therefore, even if we do not observe the abnormally increased serum LCA proportion, we may have to suspect elevated hepatic LCA concentration when liver injury appears during high-dose UDCA treatment.

Mice fed with 1% (w/w) LCA show acute intrahepatic cholestasis due to the obstruction of small bile ducts by LCA crystals (bile infarcts) [34]. These mice had markedly elevated serum ALT and ALP levels, but our UDCA-treated female DKO mice showed only moderately elevated ALT and normal ALP. Thus, liver injury in our mice was milder than in the LCA-fed mice. The UDCA-treated female DKO mice showed increased hepatic expression of inflammatory cytokines, *Tnfa*, *Ccl2*, and *Tgfb1*, but *Il1b* expression was not upregulated. The inhibited upregulation of *Il1b* in UDCA-treated DKO mice may be due to the increased hepatic concentrations of 25-HC (Fig 6B). It was reported that 25-HC synthesized by cholesterol 25-hydroxylase (CH25H) suppressed *Il1b* mRNA, protein expression, and inflammasome activity [35]. Although we did not measure the expression of *Ch25h*, another 25-HC producing enzyme, *Cyp3a11* [36], was significantly upregulated in these mice.

## Conclusion

UDCA administration to DKO mice significantly improved fertility and post-weaning mortality, presumably by making serum BAs more hydrophilic. UDCA treatment also prevented liver injury in both sexes of young DKO mice but caused hepatotoxicity in older female mice due to the accumulation of hydrophobic TLCA in the liver. These data suggest that the improvement of BA hydrophobicity by UDCA administration is sex-, age-, and organ-dependent. We must be aware of these pitfalls when using UDCA in DKO mice and possibly in humans.

## Supporting information

**S1 Checklist. The ARRIVE guidelines 2.0: Author checklist.**
(PDF)

**S1 Fig. Liver histopathology of UDCA-treated mice at a 6-week-old age.** Both male and female UDCA-treated 6-week-old DKO mice showed no significant histological findings. Hematoxylin/eosin stain. Scale bars, 100 μm. UDCA (+), with UDCA.
(TIF)

**S2 Fig. Effects of UDCA discontinuation on BA composition and liver injury.** DKO mice were bred under UDCA administration, and offspring in both sexes stopped receiving UDCA at 6 weeks old. The changes in biliary BA compositions and serum ALT activities after discontinuation of UDCA are shown. Data represent the means of duplicate determinations. The broken line indicates the boundary between conjugated and unconjugated BAs.
(TIF)

**S3 Fig. Immunohistochemical presentation of ductular reaction by immunohistochemical staining CK19.** Expression of CK19 in the livers of DKO mice at the age of 6 weeks with (+) UDCA and 20 weeks with (+) or without (-) UDCA treatment. Red arrows indicate ductular reaction by immunohistochemical staining CK19. Scale bars, 100 μm.
(TIF)

**S4 Fig. Immunohistochemistry and mRNA expression levels for macrophages.** (A) Expression of F4/80 in the livers of DKO mice at the age of 6 weeks with (+) UDCA and 20 weeks with (+) or without (-) UDCA treatment. Red arrows indicate infiltration of macrophages by immunohistochemical staining F4/80. Scale bars, 100 μm. (B) Hepatic mRNA expression levels of *Cd68* and *Cd163* in DKO mice at 20 weeks with or without UDCA treatment. Each column and error bar represents the mean and SEM. *P < 0.05, **P < 0.01, and ***P < 0.001 were considered statistically significantly different by the Tukey-Kramer test.
(TIF)

**S5 Fig. Immunohistochemistry and mRNA expression levels for neutrophils and T cells.** (A) Expression of MPO in the liver of female DKO mice at the age of 20 weeks with (+) UDCA treatment. Red arrows indicate infiltration of neutrophils by immunohistochemical staining MPO. Scale bars, 100 μm. (B) Hepatic mRNA expression levels of *Cd4, Cd8a, and Mpo* in DKO mice at 20 weeks with (+) or without (-) UDCA treatment. Each column and error bar represents the mean and SEM. ***P < 0.001 was considered statistically significantly different by the Tukey-Kramer test.
(TIF)

**S6 Fig. Hepatic mRNA expression levels of basolateral BA efflux transporters.** mRNA expression levels of *Mrp3, Mrp4, and Ostb* in the liver of DKO mice at the age of 20 weeks with

(+) or without (-) UDCA treatment. Each column and error bar represents the mean and SEM.
(TIF)

**S1 Table. Sequences of oligonucleotide primers for qRT-PCR.**
(DOCX)

**S2 Table. Effects of UDCA treatment on BA composition of total BA pool.**
(DOCX)

**S3 Table. Effects of UDCA treatment on BA concentration of total BA pool.**
(DOCX)

**S4 Table. Effects of UDCA treatment on fecal BA composition.**
(DOCX)

**S5 Table. Effects of UDCA treatment on fecal BA concentration.**
(DOCX)

**S6 Table. Effects of UDCA treatment on hepatic BA composition.**
(DOCX)

**S7 Table. Effects of UDCA treatment on hepatic BA concentration.**
(DOCX)

**S8 Table. Effects of UDCA treatment on biliary BA composition.**
(DOCX)

**S9 Table. Effects of UDCA treatment on biliary BA concentration.**
(DOCX)

**S10 Table. Effects of UDCA treatment on serum BA composition.**
(DOCX)

**S11 Table. Effects of UDCA treatment on serum BA concentration.**
(DOCX)

**S1 Data.**
(XLSX)

## Acknowledgments

We thank Hideto Takahashi, Raku Kato, Shiro Uehara, and Takahiro Machiura (The Jackson Laboratory Japan, Inc.) for the breeding and care of the mice.

## Author Contributions

**Conceptualization:** Hajime Ueda, Akira Honda.

**Data curation:** Hajime Ueda.

**Formal analysis:** Hajime Ueda.

**Funding acquisition:** Akira Honda, Takeshi Hirayama, Junichi Iwamoto, Nobuhiro Naka-moto, Tadashi Ikegami.

**Investigation:** Hajime Ueda, Akira Honda, Teruo Miyazaki, Yukio Morishita.

**Methodology:** Akira Honda, Teruo Miyazaki.

**Visualization:** Hajime Ueda, Teruo Miyazaki.

**Writing – original draft:** Hajime Ueda, Akira Honda.

**Writing – review & editing:** Teruo Miyazaki, Nobuhiro Nakamoto, Tadashi Ikegami.

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
