## [Decision Letter · Decision Letter 0]

8 Mar 2022

PONE-D-22-05356Sex-, age-, and organ-dependent improvement of bile acid hydrophobicity by ursodeoxycholic acid treatment: A study using a mouse model with human-like bile acid compositionPLOS ONE

Dear Dr. Akira Honda,

Thank you for submitting your manuscript to PLOS ONE. After careful consideration, we feel that it has merit but does not fully meet PLOS ONE’s publication criteria as it currently stands. Therefore, we invite you to submit a revised version of the manuscript that addresses the points raised during the review process.  The study has merit and we encourage resubmission.

Please submit your revised manuscript within 60 days. If you will need more time than this to complete your revisions, please reply to this message or contact the journal office at plosone@plos.org. Please include the following items when submitting your revised manuscript:A rebuttal letter that responds to each point raised by the academic editor and reviewer(s). You should upload this letter as a separate file labeled 'Response to Reviewers'.A marked-up copy of your manuscript that highlights changes made to the original version. You should upload this as a separate file labeled 'Revised Manuscript with Track Changes'.An unmarked version of your revised paper without tracked changes. You should upload this as a separate file labeled 'Manuscript'.

We look forward to receiving your revised manuscript.

Kind regards,

Gianfranco D. Alpini

Academic Editor

PLOS ONE

Journal Requirements:

2. As part of your revision, please complete and submit a copy of the Full ARRIVE 2.0 Guidelines checklist, a document that aims to improve experimental reporting and reproducibility of animal studies for purposes of post-publication data analysis and reproducibility: https://arriveguidelines.org/sites/arrive/files/Author%20Checklist%20-%20Full.pdf (PDF). Please include your completed checklist as a Supporting Information file. Note that if your paper is accepted for publication, this checklist will be published as part of your article.

Reviewers' comments:

Reviewer's Responses to Questions

**Comments to the Author**

1. Is the manuscript technically sound, and do the data support the conclusions?

Reviewer #1: Yes

Reviewer #2: Yes

2. Has the statistical analysis been performed appropriately and rigorously? 

Reviewer #1: Yes

Reviewer #2: No

3. Have the authors made all data underlying the findings in their manuscript fully available?

Reviewer #1: Yes

Reviewer #2: Yes

4. Is the manuscript presented in an intelligible fashion and written in standard English?

Reviewer #1: Yes

Reviewer #2: Yes

5. Review Comments to the Author

Reviewer #1: This is a very interesting study by Ueda et al. In this study, using a novel and recently developed human bile acid mouse mode, the Cyp2a12-/-Cyp2c70-/- double knockout (DKO) mice, it is reported that treatment with UDCA markedly improved or prevented abnormal delivery in pregnant DKO mice. However, there were severe liver injury in female, but not male UDCA, treated DKO mice. It was shown that LCA levels were higher in female DKO livers. Other studies conducted did not show a clear mechanism responsible for increased LCA in female mouse livers, including phase I metabolism or cholesterol homeostasis.

This study is very significant and comprehensive, as the new DKO model provides a new animal model with human bile acid profile to study the role of bile acids in the development of human liver diseases. Furthermore, the development of this model brings many questions that remain unanswered, which are very important to the bile acid field. There is one area in which the authors may consider is the degree of cholestasis in the female DKO mice over time, this can be done by determine the expression of several genes that are known to be altered including Mrp3 Mrp4, as well as Ostb in the liver (although the group determined CAR activity, but only narrowly). The group did determine Cyp7a1 and Cyp8b1 gene expression that were suppressed in the female mice under UDCA treatment. This is because it is possible that in female DKO mice, treatment of UDCA may exacerbate liver injury if degree of cholestasis has been more severe. Furthermore, the role of intestine FXR pathway could be determined, as UDCA is shown a FXR antagonist in the gut, which may reduce Fgf15 signaling. However, during severe cholestasis, inflammation may be more dominate than FXR pathways in the regulation of bile acid synthesis.

Reviewer #2: The authors present an interesting study on the effect of UDCA on fertility and secondary BA circulation in male and female mice at 6 and 20 week time points. The following edits are recommended for consideration of publication.

Comments:

1. More details describing the figures and tables should be included in the legends. For example, Figure 1 has thick and thin arrows but the legend does not describe what these mean. In Table 1 there are (+) and (-), but there is no indication as to what this could mean, WT and KO?

2. Isolation of microsomes should be more detailed, this method is unusual/uncommon and would be beneficial for readers to understand how it was achieved, briefly.

3. Tukey-kramer and Fisher are post hoc tests, meaning that an initial test (multiple groups = ANOVA) would have to be performed. The authors should review the statistical analysis methods and ensure clear indication of analysis performed in this manuscript.

4. Histological damage should be clearly marked with arrows, stars, arrowheads, etc. Further, ductular reaction and immune cell infiltration should be stained for and quantified with IHC for CK-19 (bile duct mass), F4/80 or CD68 (macrophage marker), and CD4/8 for T cells, and MPO for monocytes. These efforts will increase understanding and impact to the readers of the results determined by the authors.

5. The increased fecal excretion and pool of TBA indicate that UDCA administration in Female DKO mice may increase BA production. The authors should focus on investigating the role of CYP genes responsible for BA synthesis and the role of ASBT in the intestine of these mice (since small intestine was collected). These studies along with estrogen signaling may allow the authors to increase the impact and novel approach to this study.

6. It is alarming that UDCA treatment increased inflammation (seen in PCR) in the liver of female DKO. Does this indicate that the benefits of UDCA (lower miscarriage) comes at the cost of mother hepatic injury?

7. The discussion needs to be shortened to succinctly indicate where this study stands in the current literature. While it is important to emphasize key findings of this study, continuous confirmation of the previous DKO study dampens these data impact. It is recommended that the authors increase citation of works on BAs in pregnant women to identify any connections between this work and that of pregnant cholestasis.

6. PLOS authors have the option to publish the peer review history of their article (what does this mean?). If published, this will include your full peer review and any attached files.

Reviewer #1: **Yes: **Grace Guo

Reviewer #2: **Yes: **Vik Meadows

---

## [Author Response · Author response to Decision Letter 0]

3 Jun 2022

Replies to the comments from PLOS ONE reviewers:

Reviewer #1:

This study is very significant and comprehensive, as the new DKO model provides a new animal model with human bile acid profile to study the role of bile acids in the development of human liver diseases. Furthermore, the development of this model brings many questions that remain unanswered, which are very important to the bile acid field. There is one area in which the authors may consider is the degree of cholestasis in the female DKO mice over time, this can be done by determine the expression of several genes that are known to be altered including Mrp3 Mrp4, as well as Ostb in the liver (although the group determined CAR activity, but only narrowly). The group did determine Cyp7a1 and Cyp8b1 gene expression that were suppressed in the female mice under UDCA treatment. This is because it is possible that in female DKO mice, treatment of UDCA may exacerbate liver injury if degree of cholestasis has been more severe. Furthermore, the role of intestine FXR pathway could be determined, as UDCA is shown a FXR antagonist in the gut, which may reduce Fgf15 signaling. However, during severe cholestasis, inflammation may be more dominate than FXR pathways in the regulation of bile acid synthesis.

RESPONSE: Thank you for the reviewer’s comments. We think hepatic BA concentration is the best marker to assess a degree of cholestasis. Serum ALP activity is a useful surrogate marker for cholestasis in humans, but it is not very sensitive in mice because main mouse serum ALP isoenzyme is bone-derived one (Hatayama et al. J. Toxicol. Sci. 36:211-219, 2011). As suggested by the reviewer, we have measured hepatic mRNA expression levels of Mrp3, Mrp4, and Ostb and added them to S6 Fig. UDCA treatment tended to upregulate Mrp3 and Mrp4 expressions (Fxr independent) in both sexes, but the differences were not statistically significant. The expression levels of Ostb (FXR dependent) in females tended to be higher than those in males, but UDCA did not change the expressions significantly in both sexes. Thus, although hepatic BA concentrations, especially TLCA, were markedly elevated in female UDCA-treated DKO mice, significant upregulations of these transporters were not observed. The results suggest that TLCA may not be an effective activator of these transporters, or inflammation may be more dominant than FXR pathways in the regulation of BA transport as well as BA synthesis. We have added some discussion on the basolateral BA efflux transporters (lines 430-434).

Reviewer #2:

1. More details describing the figures and tables should be included in the legends. For example, Figure 1 has thick and thin arrows but the legend does not describe what these mean. In Table 1 there are (+) and (-), but there is no indication as to what this could mean, WT and KO?

RESPONSE: Thank you for the reviewer’s suggestions. We have revised legends of figures and tables to describe more detailed information. In Figure 1, thick and thin arrows mean the enterohepatic circulation of LCA is usually less (thin arrows) than those of other BAs (thick arrows). In Table 1, (+) and (-) mean pregnancy and no pregnancy. In Figures 2-6, (+) and (-) mean with and without UDCA treatment.

2. Isolation of microsomes should be more detailed, this method is unusual/uncommon and would be beneficial for readers to understand how it was achieved, briefly.

RESPONSE: We have added more detailed microsome preparation method to Materials and methods section (lines 155-161).

3. Tukey-kramer and Fisher are post hoc tests, meaning that an initial test (multiple groups = ANOVA) would have to be performed. The authors should review the statistical analysis methods and ensure clear indication of analysis performed in this manuscript.

RESPONSE: We agree to the reviewer’s comment, but our Fisher is not Fisher’s least significant difference test but Fisher’s exact probability test. We used the Tukey–Kramer test after one-way ANOVA but did not mention it in the text. We have revised the Statistics section in Materials and Methods (lines 178-181). 

4. Histological damage should be clearly marked with arrows, stars, arrowheads, etc. Further, ductular reaction and immune cell infiltration should be stained for and quantified with IHC for CK-19 (bile duct mass), F4/80 or CD68 (macrophage marker), and CD4/8 for T cells, and MPO for monocytes. These efforts will increase understanding and impact to the readers of the results determined by the authors.

RESPONSE: Thank you for the reviewer’s valuable comments. We have marked histological damages with arrows in Figure 2B. We have confirmed the proliferation of bile ductular structure by IHC stain of CK19 (S3 Fig), infiltration of macrophages by IHC stain of F4/80 and mRNA expression levels of Cd68 and Cd163 (S4 Fig), and that of neutrophils and T cells by IHC stain of MPO and mRNA expression levels of Mpo, Cd4, and Cd8 (S5 Fig). We have added this information to the Result section (lines 214-217).

5. The increased fecal excretion and pool of TBA indicate that UDCA administration in Female DKO mice may increase BA production. The authors should focus on investigating the role of CYP genes responsible for BA synthesis and the role of ASBT in the intestine of these mice (since small intestine was collected). These studies along with estrogen signaling may allow the authors to increase the impact and novel approach to this study.

RESPONSE: UDCA administration markedly increased the proportion of UDCA with a considerable reduction of CA and CDCA in the BA pool of female DKO mice. Since UDCA is not a central primary BA in mice, the increase in the BA pool in UDCA treated female DKO mice is likely due to UDCA ingestion itself, not increased BA production. Female DKO mice had larger BA pools than male DKO mice regardless of UDCA administration. We compared CYP7A1 activity and Cyp27a1 expression between the sexes (Fig.6D), but there was no significant increase in either enzyme in females. We collected the small intestines in this study but used them all to measure the BA pool. We previously measured ileal Asbt expressions in DKO mice (Ref. #8), and we found no significant difference between the sexes. In addition, unlike mice, males have larger BA pools than females in humans. Thus, the mechanism of the sex difference in the BA pool size remains unknown and is not simply an effect of estrogen.

6. It is alarming that UDCA treatment increased inflammation (seen in PCR) in the liver of female DKO. Does this indicate that the benefits of UDCA (lower miscarriage) comes at the cost of mother hepatic injury?

RESPONSE: Thank you for the reviewer’s question. Mothers do not have a hepatic injury since matings are made at nine weeks of age in our usual experiments. More prolonged UDCA treatment (20 weeks) causes liver injury in female DKO mice. If UDCA is administered only during pregnancy, older mice may be able to give birth normally without liver damage. However, we have never done that.

7. The discussion needs to be shortened to succinctly indicate where this study stands in the current literature. While it is important to emphasize key findings of this study, continuous confirmation of the previous DKO study dampens these data impact. It is recommended that the authors increase citation of works on BAs in pregnant women to identify any connections between this work and that of pregnant cholestasis.

RESPONSE: Thank you for the reviewer’s comments. We have deleted the discussion part that continuously confirms the previous Cyp2c70 KO study (between line 402 and line 403). As suggested by the reviewer, the connection between pregnant DKO mice and intrahepatic cholestasis of pregnancy (ICP) may be interesting. Increased serum hydrophobic BAs (due to altered BA composition or cholestasis) and frequent premature delivery are common features (Ref. #19). However, UDCA treatment prevented premature delivery in DKO mice while it was ineffective in reducing adverse perinatal outcomes in patients with ICP (Ref. #22). Therefore, it is unclear whether pregnant DKO mice would be a good model for ICP. We have added these considerations and references to the Discussion (lines 374-377 and 384-386).

---

## [Decision Letter · Decision Letter 1]

13 Jun 2022

PONE-D-22-05356R1Sex-, age-, and organ-dependent improvement of bile acid hydrophobicity by ursodeoxycholic acid treatment: A study using a mouse model with human-like bile acid compositionPLOS ONE

Dear Dr. Akira Honda,

Thank you for submitting your manuscript to PLOS ONE. After careful consideration, we feel that it has merit but does not fully meet PLOS ONE’s publication criteria as it currently stands. Therefore, we invite you to submit a revised version of the manuscript that addresses the points raised during the review process.  Please address a few minor comments.

Please submit your revised manuscript within 30 days. If you will need more time than this to complete your revisions, please reply to this message or contact the journal office at plosone@plos.org. Please include the following items when submitting your revised manuscript:A rebuttal letter that responds to each point raised by the academic editor and reviewer(s). You should upload this letter as a separate file labeled 'Response to Reviewers'.A marked-up copy of your manuscript that highlights changes made to the original version. You should upload this as a separate file labeled 'Revised Manuscript with Track Changes'.An unmarked version of your revised paper without tracked changes. You should upload this as a separate file labeled 'Manuscript'.If applicable, we recommend that you deposit your laboratory protocols in protocols.io to enhance the reproducibility of your results. Protocols.io assigns your protocol its own identifier (DOI) so that it can be cited independently in the future. For instructions see: https://journals.plos.org/plosone/s/submission-guidelines#loc-laboratory-protocols. Additionally, PLOS ONE offers an option for publishing peer-reviewed Lab Protocol articles, which describe protocols hosted on protocols.io. Read more information on sharing protocols at https://plos.org/protocols?utm_medium=editorial-email&utm_source=authorletters&utm_campaign=protocols.

We look forward to receiving your revised manuscript.

Kind regards,

Gianfranco D. Alpini

Academic Editor

PLOS ONE

Journal Requirements:

Reviewers' comments:

Reviewer's Responses to Questions

**Comments to the Author**

1. If the authors have adequately addressed your comments raised in a previous round of review and you feel that this manuscript is now acceptable for publication, you may indicate that here to bypass the “Comments to the Author” section, enter your conflict of interest statement in the “Confidential to Editor” section, and submit your "Accept" recommendation.

Reviewer #1: All comments have been addressed

Reviewer #2: All comments have been addressed

2. Is the manuscript technically sound, and do the data support the conclusions?

Reviewer #1: Yes

Reviewer #2: Partly

3. Has the statistical analysis been performed appropriately and rigorously? 

Reviewer #1: Yes

Reviewer #2: Yes

4. Have the authors made all data underlying the findings in their manuscript fully available?

Reviewer #1: Yes

Reviewer #2: No

5. Is the manuscript presented in an intelligible fashion and written in standard English?

Reviewer #1: Yes

Reviewer #2: Yes

6. Review Comments to the Author

Reviewer #1: The concerns were well addressed. The author have done a good job addressing the raised concerns. Further experimental data were supplied to support the conclusion.

Reviewer #2: The authors have addressed reviewer comments, but the CK-19 staining does not stain bile ducts. The arrows seem to indicate focal macrophage clusters instead of bile ducts. It is apparent in all images since CK-19 does not demark the bile ducts near portal veins. I would suggest that the authors review this manuscript here (PMID: 32961356) and obtain more specific CK-19 antibodies that will allow them to stain and quantify the CK-19 staining to measure bile duct mass/ductular reaction. CK-19 positivity should not exist in the sinusoidal space or within the parenchyma as it is only expressed by cholangiocytes in the liver.

7. PLOS authors have the option to publish the peer review history of their article (what does this mean?). If published, this will include your full peer review and any attached files.

Reviewer #1: No

Reviewer #2: **Yes: **Vik Meadows

---

## [Author Response · Author response to Decision Letter 1]

25 Jun 2022

Replies to the comment from Reviewer #2:

The authors have addressed reviewer comments, but the CK-19 staining does not stain bile ducts. The arrows seem to indicate focal macrophage clusters instead of bile ducts. It is apparent in all images since CK-19 does not demark the bile ducts near portal veins. I would suggest that the authors review this manuscript here (PMID: 32961356) and obtain more specific CK-19 antibodies that will allow them to stain and quantify the CK-19 staining to measure bile duct mass/ductular reaction. CK-19 positivity should not exist in the sinusoidal space or within the parenchyma as it is only expressed by cholangiocytes in the liver.

RESPONSE: Thank you very much for the reviewer’s valuable comments and suggestions. We performed CK-19 staining using different mouse IgG blocking reagents and obtained better results. We have replaced the previous pictures with better ones (S3 Fig) and revised the Histopathological examination section in the manuscript (lines 133-137). In addition to CK-19, we again stained F4/80 (S4 Fig) and MPO (S5 Fig) with better conditions and replaced the previous pictures with new ones.

---

## [Editor Report · Decision Letter 2]

28 Jun 2022

Sex-, age-, and organ-dependent improvement of bile acid hydrophobicity by ursodeoxycholic acid treatment: A study using a mouse model with human-like bile acid composition

PONE-D-22-05356R2

Dear Dr. Honda,

We’re pleased to inform you that your manuscript has been judged scientifically suitable for publication and will be formally accepted for publication once it meets all outstanding technical requirements.

Kind regards,

Gianfranco D. Alpini

Academic Editor

PLOS ONE
---

## [Editor Report · Acceptance letter]

4 Jul 2022

PONE-D-22-05356R2 

Sex-, age-, and organ-dependent improvement of bile acid hydrophobicity by ursodeoxycholic acid treatment: A study using a mouse model with human-like bile acid composition 

Dear Dr. Honda:

I'm pleased to inform you that your manuscript has been deemed suitable for publication in PLOS ONE. Congratulations! Your manuscript is now with our production department. 

Kind regards, 

on behalf of

Dr. Gianfranco D. Alpini 

Academic Editor

PLOS ONE